# Ecological Effect of Ecological Engineering Projects on Low-Temperature Forest Cover in Great Khingan Mountain, China

**DOI:** 10.3390/ijerph182010625

**Published:** 2021-10-11

**Authors:** Shuqing Wang, Run Zhong, Lin Liu, Jianjun Zhang

**Affiliations:** 1School of Land Science and Technology, China University of Geosciences (Beijing), Beijing 100083, China; wangsq@cugb.edu.cn (S.W.); z_xb_2003@163.com (R.Z.); zhangjianjun_bj@126.com (J.Z.); 2School of Land Science and Space Planning, Hebei GEO University, Shijiazhuang 050031, China

**Keywords:** SDG indicator, forest degradation, land productivity, land degradation

## Abstract

The evaluation of ecological restoration projects can provide support for further strengthening the efforts of ecological restoration work and implementing the strategic objectives of the ecological region. Considering the current problem of the single evaluation index, this study evaluated the implementation effect of ecological projects from different temporal and spatial dimensions. Based on the MODIS vegetation index time series data, this study first computed the Sustainable Development Goal (SDG) indicator 15.3.1 of Great Khingan Mountain (GKM) to evaluate the impact of ecological engineering on land use change and land productivity. As a common indicator, the Normalized Difference Vegetation Index (NDVI) values showed a trend of a decrease and then gradual increase after the start of the Natural Forest Protection Project (NFPP) II, which was related to the land use changes from the forest to the grassland during the implementation of the NFPP. However, land productivity maintained a steady trend because of the transition between the forest and grassland. Meanwhile, to detect changes in vegetation at a smaller scale, the LandTrendr algorithm was used to identify the magnitude of forest disturbance, the years when it occurred, and the year of restoration. After implementing the ecological project, the forests in the GKM region were only partially disturbed, and most of the forests in most areas maintained a stable trend. Our study highlighted the varying effectiveness of different indexes for NFPP and evaluated the ecological impact of ecological projects from multiple perspectives.

## 1. Introduction

Ecosystems are mainly influenced by human activities and climate variations [1]. With the development of the global modernization trend, a larger scale of environmental pollution and destruction is accompanied by a new wave of worldwide environmental protection. Many large-scale ecological restoration projects, especially forest restoration projects were underway for more than 20 years to reverse ecological degradation and achieve environmental sustainability [2]. Of these ecological projects, forest conservation and restoration projects are the most important, as the effects of forest loss and degradation are felt on all scales, from global climate change to the decline in the economic value of forest resources and biodiversity, and threats to local livelihoods [3,4]. The New Zealand government announced the One Billion Trees (1 BT) program which aimed to plant one billion trees in the country by 2028 [3,5]. Reducing Emissions from Deforestation and Forest Degradation plus (REDD+), as the best-known international forestry-based policy for carbon dioxide removal, was created by the United Nations Framework Convention on Climate Change (UNFCCC) Conference and aimed to implement schemes by national governments to reduce human impact on forests, an activity which results in greenhouse gas emissions at the national level [6]. 

China has a long history of forest degradation and restoration. To minimize the human impact on the environment and restore vegetation coverage, a series of vegetation-related policies were implemented by China [7]. The largest project is the Three-North Shelter Forest Program (TNSFP), focused on increasing forest cover from 5% to 15% from 1978 to 2050 [8]. In 2000, the Chinese government fully implemented the Natural Forest Protection Project (NFPP), which instilled logging bans and harvesting reductions in 68.2 million ha of forest land. The main goal of this project was to protect the existing natural forests, increase vegetation cover and mitigate soil erosion and desertification through the revegetation and conversion of agricultural land to forest land [9]. In 2014, the State Forestry Administration (SFA) expanded the scope of NFPP and tried to ban commercial logging in the state-owned natural forests of Heilongjiang Province. The log supply in Heilongjiang Province historically accounted for more than 30% of China’s domestic log supply.

A large number of ecological projects were implemented and estimating the effects of these ecological projects became an important research direction. Several studies investigated the spatial and temporal effects of climatic factors and human activities on changes in vegetation productivity from 1985 to 2015, where actual net primary productivity (ANPP) and net primary productivity (NPP) were often used to quantify vegetation dynamics [3,7]. Several studies used multiple satellite images to assess the changes of the leaf area index (LAI), gross primary productivity (GPP), Normalized Difference Vegetation Index (NDVI), and aboveground biomass in different regions in China [10,11]. Previous studies focused on long-term change analysis, often failing to capture such subtle changes which occurred over long periods, but could have significant impacts on forest structure, composition, and function, and thus ultimately limited the successful implementation of sustainable development goals (SDG) [12].

Trend and change analysis using different time series data is the most common method when performing calculations using the different indexes. To find the trend of NDVI or GPP time series, the linear regression technique is a viable and robust method [13]. As vegetation change is influenced not only by human activities but also by the climate, a residual trend analysis is now commonly used to separate human-induced and climate-driven vegetation changes [14,15]. The Breaks for Additive Season and Trend (BFAST) algorithm and LandTrendr algorithm are used to identify long-term trends and abrupt changes (breaks) in the time series [16,17,18]. These methods use multivariate (parametric) time series analysis and effective algorithms to comprehensively assess the effects of vegetation restoration, which provide us with the data, information, and knowledge necessary to better implement and manage large-scale ecological projects. Although all of the above methods are used in many current studies, there is relatively little research on applying them to analyze the impact of ecological engineering on the environment.

To integrate the application of different methods to evaluate the impact of ecological engineering, this study uses the alpine region of Great Khingan Mountain (GKM) as the research object and integrates the application of various methods. In this study design, the three sub-indicators used for SDG 15.3.1 are first used to measure different aspects of land cover which relate to vegetation. Primary productivity, the first sub-indicator, can directly measure changes in the biomass present in an area, but it is not able to determine whether this change is positive, as not all increases in plant biomass are interpreted as improvements. As the second sub-indicator, land cover fills this gap. It explains the landscape from a thematic point of view, looking at the features that were there previously and those that are there now. It includes changes in vegetation, as well as bare land, cities, and water. The last sub-indicator, the Soil Organic Carbon Index, uses a land cover map to inform changes in the organic carbon of the soil over time. Although the results of this method are not ideal, considering the status of global soil science and measurement, researchers agree that this method is the best method to implement on a global scale. Vegetation is a key component of most ecosystems and a good representation of its overall function and health. For example, within a particular land cover type, the land productivity or soil carbon levels may change over time. However, changes in land cover types usually lead to changes in the level and dynamics of land productivity, which in turn affect the carbon storage of a particular area. Therefore, when evaluating the degradation status against these three sub-indices, it is convenient to summarize the fine-scale results (such as the pixel-scale evaluation in this method) into the spatial characteristics determined in the land cover and land cover change sub-indices. In addition to the SDG factor, in order to find changes, disturbances, and causes of vegetation at a smaller scale, the LandTrendr algorithm is used in this study to identify disturbances in the vegetation of the study area.

Forest change can be influenced by many factors. In this study, the influence of other climatic factors are removed from the calculations to study the impact of ecological projects. This means that the study assumes that external factors such as the climate are consistent over the period of forest change in the study area. To assess the interest and potential of ecological engineering in forests from different perspectives, this study uses images at different scales to identify the forest degradation and improvement in order to explore the possible impacts of different contexts and policies. 

## 2. Study Area and Data

### 2.1. Study Area

As the northernmost border area in China, the Greater Khingan Mountains are located at 121°12′−127°00′ East longitude and 50°10′−53°33′ North latitude. With an average altitude of 573 meters, it has a cold temperate and a continental monsoon climate with short summers and long winters. The annual average frost-free period is only 80−110 days. The annual average temperature is −2.1 °C, and the historical minimum temperature is −52.3 °C. In the study area, the mountains are mainly covered by boreal coniferous forest and grass (Figure 1). The huge mountains and forests effectively block the Siberian cold current and the cold wind of the Mongolian plateau and become a natural barrier for the Songnen Plain in northeast China and the Hulunbuir grassland in inner Mongolia. This area is also an important water conservation area for the Heilongjiang and Nenjiang rivers and other water sources, guaranteeing the water supply for residents in the middle and lower reaches of the two rivers.

### 2.2. Data

#### 2.2.1. NDVI Datasets

Normalized Difference Vegetation Index (NDVI) (Exelis Visual Information Solutions, Boulder, CO, USA) is often used to measure vegetation cover and health. Moderate Resolution Imaging Spectroradiometer (MODIS) and Landsat provide two-scale NDVI data sets. MODIS products provide vegetation indices with 16-day intervals and multiple spatial resolutions, providing a consistent spatial and temporal comparison of vegetation canopy greenness, which is a comprehensive attribute of leaf area, chlorophyll, and canopy structure. The MODIS Vegetation Indices product, MOD13Q1, at 250 m spatial resolution, was selected to compute the annual value and land productivity, as well as degradation. Landsat data sets provided the Surface Reflectance-derived NDVI with a 30-meter resolution, derived from Landsat 4–5 Thematic Mapper (TM), Landsat 7 Enhanced Thematic Mapper Plus (ETM+), and Landsat 8 Operational Land Image (OLI)/Thermal Infrared Sensor (TIRS). The finer resolution NDVI dataset was used to quantify vegetation greenness and was more helpful to assess changes in plant health. The data used in this study are shown in Table 1.

#### 2.2.2. Land Cover and Climate Data

To assess changes in land cover, we used land cover maps downloaded from ESA CCI (European Space Agency Climate Change Initiative) datasets, which covering the study area for the baseline and target years. The land use classes included forest, grassland, cropland, wetland, artificial areas, bare areas and water, with a 300 m resolution.

Land productivity was affected by several factors, such as temperature, the availability of light, nutrients, and water. Water availability had a significant influence on the amount of plant tissue produced every year. It was important to interpret the results in the context of the information on historical precipitation. When the land trends were driven by regional patterns of changes in water availability, the declining productivity of these trends could be identified as human-caused land degradation. The precipitation product CHIRPS (Climate Hazards Group InfraRed Precipitation) was used to eliminate the effects of the climate. From 1981 to the present, CHIRPS combined our internal climatology, CHPclim, 0.05° resolution satellite imagery, and in situ station data to create a gridded rainfall time series for trend analysis and seasonal drought monitoring.

## 3. Method

The effect of ecological engineering projects policies on forests is a long-term process, so time series analyses with multivariable (parameters) and effective algorithms are necessary to comprehensively assess the effectiveness of vegetation restoration which provide essential data, information, and knowledge. Due to the long duration of ecological projects and the number of influencing factors, a multi-parameter time series analysis was used to assess the impact of ecological projects in providing the necessary data and information for forest restoration. The temporal dynamics of terrestrial ecosystems commonly consisted of continuous changes, discontinuous changes, and no changes [10]. The ability to identify the trend or dynamic change in policies, not the effect of natural factors (precipitation, fire, temperature, diseases) is important for better assessing the effects of policies. In this study design, we conducted a time series analysis using multi-scale satellite images, climate data, and statistical data (Figure 2).

### 3.1. Land Productivity Index

The land productivity is the maximum productivity of the land at the current level of farming technology and the measures relevant to it. It is the source of all food, fiber, and fuel that sustains human life. As land productivity can reflect long-term changes that indicate the health and productivity of the land and the net effect of changes in the ecosystem functions of plants and biomass growth, land productivity can be used as an indicator in the evaluation of ecological projects. Land productivity can be measured over large areas by the satellite Earth observations of NPP. The productivity index is the algorithm used to measure land productivity levels from image data. This study used *NDVI* [19] as the surrogates of NPP. To separate degradation effects from other sources of variation in productivity observations, a calibration was performed by calculating the ratio of *iNDVI* to *iET* (Evapotranspiration) to minimize the influence of climatic or seasonal factors [20]. The method for calculating water use efficiency (WUE) with corrected *iNDVI* (*iNDVIw*) per year is:(1)$$iNDVIw=\frac{iNDVI}{iET}$$
where iNDVI is the *NDVI* integrated over the growing season or relevant period each year, and iET is *ET* integrated over the same period.

Productivity was assessed in terms of trajectory, performance, and status. Trajectories could be used to identify degradation in areas with increasing productivity trends, and performance could identify low productivity compared to other areas with similar land cover types and similar climatic conditions. Status was used to compare the historical range of productivity levels at the site over time. 

Productivity trajectory was calculated using the Thiel-Sen median, a robust, non-parametric linear regression method. The trend significance was determined by Mann–Kendall [21,22,23]. Positive and negative z-scores indicated trends of increasing productivity or decreasing productivity. The significance of the slope of the trajectory, calculated at the *p* = 0.05 level for more than eight data points, should be reported on three scales: improved, degraded, and insignificant.

Productivity state represented the level of productivity in a given spatial unit compared to the observed productivity levels for that spatial unit over time. Productivity state could be interpreted as an indicator of the relative standing biomass [24]. Existing degradation based on productivity performance could be identified by a mean productivity performance in the baseline period of less than 50% of the potential maximum. The iNDVIw values were classified during the baseline period into ten decile classes using the unsupervised ISODATA classification. These become the baseline iNDVIw classes. Productivity state change was assessed by comparing iNDVIw in the assessment year to the baseline iNDVIw classes. 

As the NFPP was a long-term project, this study used 2000 as a cut-off point to analyze the different land use changes and land degradation before and after the NFPP. The year 2010, the start of the second phase of the NFPP, was chosen as the interval. The average productivity of the early epoch (2001 to 2010) and late epoch (2011 to 2018) could be calculated. Then, pixels in which the productivity level decresed between early and late epochs were identified for this metric.

### 3.2. Land Trendr

To detect trends in forest disturbance and recovery at multi-scale using yearly Landsat time series, we used LandTrendr as temporal segmentation algorithms. LandTrendr reduced image data to a single band or spectral index and then divided it into a series of straight-line segments by breakpoint (vertex) identification [3,18]. In practice, LandTrendr looks at the spectral history of a pixel to identify the breakpoints which separate periods of persistent change or stability in the spectral trajectory and records the year in which the change occurred. In this study, the breakpoints, durable changes, and the year could be used to detect trends in forest disturbance and recovery. Meanwhile, the typical area could be selected and compared with the Landsat remote sensing images selected on Google Earth to explore the possible reason for disturbance. 

## 4. Results

### 4.1. Trend and Changes of Land Productivity Index

Land cover addresses the state and changes in the structure and composition of the landscape, from natural events and human activities. As the NFPP was a long-term project this study used 2000 as a cut-off point to analyze the different land use changes and land degradation before and after the NFPP. The forest was the dominant land use type in the study area throughout the project period. The western side of GKM was mainly grassland, while the eastern side was cropland (Figure 3a,b). The land cover transition map in Figure 3c showed that most land use types maintained the original state for a long time before the NFPP. After project implementation, the type of land use conversion in the area was dominated by the loss of forest (Figure 4a,b), and the distribution of forest loss was more concentrated (Figure 4c). According to the definition of land cover degradation in SDG, Figure 3d and Figure 4d showed the results of the land cover degradation in the study area. After the project implementation, there was a higher concentration of land degradation appearing in the north and northeast area and some areas of land improvement (Figure 4d). Additionally, prior to the start of the project, the study area was less prone to land degradation and land improvement (Figure 3d).

Figure 5 compared the different changes in the quantity of land cover before and after the implementation of the project. Before 2000, land use losses were dominated by forest losses. This was inseparable from the deforestation of that period. After 2000, the natural forest protection project was implemented. After the implementation of the policy, land use losses became dominated by grassland losses.

The ecological engineering projects had an effect not only on the land use change but also on forest degradation or deforestation. Figure 6 showed the trend of NDVI using the annual NDVI value. The NDVI showed a stable increase trend while experiencing a dramatic decrease from 2002 to 2003. The annual NDVI values maintained a decreasing trend under the trend line until 2010. At that time, all large-scale ecological engineering projects, especially NFPP, were launched by the Chinese Government to speed up the restoration of the forest ecosystems.

Land productivity reflected the net effects of changes in the ecosystem functioning on plant and biomass growth. Figure 7 interpretated the trend and its significance regarding the variability of annual NPP in the time series. Throughout the project period, most of the study area showed an increase in land productivity, a small amount of stability and a very small trend of degradation: 67.3%, 31.8%, and 0.9%, respectively.

Productivity state change can be reported in terms of three classes relative to the baseline productivity state: (1) Improvement: observed productivity in baseline iNDVI classes 9 or 10. (2) Stable: observed productivity in baseline iNDVI classes 6, 7 or 8. (3) Degradation: observed productivity in baseline iNDVI classes 1, 2, 3, 4 or 5. The forest experienced a degradation between the two periods, whereas the other land showed an improvement (Figure 8). The degradation areas, especially in forest area, consisted of productivity trajectory degradation. Figure 9 showed that most of the land use types had no degradation, except 0.6% grassland and cropland located in the eastern and southeastern areas.

### 4.2. Forest Change Monitoring

#### 4.2.1. Magnitude, Duration, and Years of Land Use Changes

Corse-scale image analyses revealed that the land cover/use and land productivity of the study area only had a little degradation (Figure 9). The magnitude, duration, and years of land use changes were computed by the LandTrendr algorithm shown in Figure 10. The higher disturbances of magnitude occurred only in some parts of the central area. The relatively high disturbances occurred in the northeastern area near the border. Most of the remaining areas were largely free of disturbances. The larger disturbances occurred in 2003 and 2006. The vast majority of vegetation disturbance across the study region was mostly short-term disturbance with rapid recovery, lasting up to three years (e.g., yellow areas in the northeast of the map).

Taking 2000 as the starting point of the disturbance, the obtained annual time series of the disturbed areas from 2001 to 2020 is shown in Figure 11a. It can be seen that the disturbed area in 2001 reached 330,000 ha, accounting for 35.8%, followed by larger disturbed areas in 2003, 2006, 2015, and 2020.

The study area was mainly disturbed for a short period and the disturbance durations were all within three years. This indicated that the vegetation in the Daxinganling area recovered within a relatively short time after the disturbance occurred. In this area, the disturbance duration was approximately 1–2 years, and the disturbance duration area ratio reached 72%, as shown in Figure 11b.

#### 4.2.2. Detection of Typical Deforest Area

The typical sites were selected to explore the fine-scale changes and reasons for changes using the LandTrendr algorithms. Figure 12 shows the distribution of typical area and the high-resolution images at different years. From the typical area selected in Figure 12a, it can be seen that the main disturbance in this area occurred in the year 2003, and the Landsat remote sensing images of 2004, 2005, and 2006 were selected from Google Earth for comparison in Figure 12b. Additionally, it was found that there was a certain amount of disturbance in all years in this area, but none of these disturbances were the main disturbance, and the disturbance in 2003 was the main interference, consistent with the main interference obtained by LandTrendr algorithm.

In the typical region, the sampling point (123°15′53.70′′ E, 51°26′5.26′′ N) was selected for the LandTrendr algorithm pixel time series trajectory extraction, and the time series was 2000–2020. The Normalized Burn Ratio (NBR) index (multiplied by 1000) was fitted to make the change information more intuitive, shown in Figure 13. From the index trajectory of the sampling point, the NBR value of this point ranged from 0.355 to 0.487. 

In 2003 after a serious disturbance, the NBR value was −0.118, demonstrating bare soil. The area was continuously disturbed for a short period of 1 year and started to recover gradually after 1 to 2 years. After 8 to 10 years, it returned to the initial vegetation cover level after the disturbance.

## 5. Discussion

### 5.1. Land Use Change Trend

This research compared the similarities and differences in land use changes before and after NFPP implementation, as shown in Figure 5. Until the late 1990s, forests were reported to show a trend of shrinkage [25,26]. After the implementation of the policy in 2000, the loss of forests decelerated, while the loss of grasslands became the most significant loss. The trends in forest changes were consistent with those previously reported [27]. The results of land use change were related to policies in the natural forest protection project: reducing the amount of natural forest resources harvested in state-owned forest areas in northeastern and inner Mongolia, strictly controlling timber consumption and eliminating over-limit harvesting. However, after the implementation of the natural forest protection policy, while deforestation was halted under the constraints of a strict policy, the loss of grasslands increased. The dramatic increase in population led to the occupation of grassland and the clearing of arable land and other land use types [22,28]. Meanwhile, the decline in grassland trends found in this study were consistent with those found in the other literature. The local peasants generally prefer to convert their cropland into forest rather than grassland to receive a higher compensation [29]. 

This study showed the trend of the annual mean NDVI; that the NDVI began to decline sharply from 2003 and maintained a downward trend until 2013, when an upward trend was observed. The trend in NDVI change is consistent with the trend in land use change: after the implementation of the 2000 policy, there was also a loss of forests, especially grasslands, which led to a decrease in the annual average NDVI value [30]. In addition, the forest gain after 2004 was in line with investments at the early stages of the ecological programs [31].

### 5.2. Land Productivity Trend

In addition to the first indication of land cover change, representing to some extent the underlying use, as well as land conversion and the resulting habitat fragmentation, land productivity provided an indication of ecosystem function and health and brought increased attention to ecosystem services. On average, land productivity in GKM maintained a stable state throughout the implementation of the NFPP. The improvement in trajectory was seen in grasslands and croplands (on the sides of the forest in Figure 9). Most of the forest remained intact, with a small proportion degraded to some extent. Land productivity trends showed that this research was consistent with the results of other methods and field surveys [32]. 

From 2010, the government began implementing the second phase of the NFPP in key state-owned forest areas in northeast China and inner Mongolia [31]. The aim of the second phase of the project was to continue to protect the natural forest and, at the same time, to treat the previously cultivated tree species on a merit basis [22,33,34]. In this study, land productivity degradation was calculated using 2010 as the time transition point (shown in Figure 8). After 2010, there was a certain degree of productivity degradation in the forested part of GKM, which was related to the purpose of the NFPP II. In Phase II, there were many low-yielding and inefficient forests that needed to be renovated and nurtured. Additionally, natural forests only recently entered the stage of recovery and development, and the quality of forest resources remains insufficient, with a large proportion of medium and young forests. Moreover, within the NFPP provinces, Heilongjiang and Inner Mongolia in northeastern China, were designated in the NFPP as “key state-owned forest regions.” Logging was not as strictly banned in this region as it was in the watersheds of the Yangtze and Yellow Rivers [35]. So, in the forest core of GKM, there was a land productivity degradation, due to the felling of weak and ecologically worthless tree species and deforestation. In 2014, the State Forestry Administration (SFA) expanded the NFPP with the launch of a trial ban on commercial logging in state-owned natural forests in Heilongjiang Province [33]. The NDVI trends shown in Figure 6 were consistent with the new commercial logging ban.

This study used SDG 15.3.1 as an important indicator to evaluate the effectiveness of large-scale ecological projects, with three sub-objectives in land productivity trajectory degradation, state degradation and performance degradation. The results were consistent with the situation of actual land use change and policy direction. The sub-indicator based on SDG 15.3.1 provided a reliable and practical method to assess the effect of ecological projects. 

### 5.3. LandTrendr Analyses

The LandTrendr algorithm detected the magnitude of vegetation change, the year of change, and the timing of recovery at a more granular level. In terms of the magnitude of disturbance occurrence, there was a concentration of disturbances in the forest area, mainly in three areas: the Genhe area in the central part of the forest area and the Mohe area in the northeast. The disturbances occurred at different times in the three areas, in 2003, 2006, and 2010. The magnitude of disturbance was moderate and light, except in the Genhe area. To further identify the causes of disturbance in the Genhe area, the high-resolution images of the surrounding years in which the disturbances occurred are given in Figure 11. It can be seen from the images that a dramatic change occurred in the Genhe region in 2003. The identification of the images and the comparison of the information shows that a forest fire occurred in the area in 2003 [36]. Following the forest fires, the forest gradually recovered. Compared to other disturbed areas, the recovery period in this area was longer: around 3 years. Since 2008 there was a decreasing trend in forest disturbance. The implementation of the "NFPP II" campaign for nature reserves, in 2011, with the Ministry of Environmental Protection as the main enforcement agency, served as a catalyst for the protection and supervision of nature reserves at the provincial and municipal levels. 

The distribution of disturbance outside the forest area was more dispersed, mainly on grassland and cultivated land to the east and west of the forest area. Disturbances occurred to a lesser extent and largely occurred around 2004. In terms of the impact of policy, the implementation of strict natural forest protection projects after 2000 did effectively protect the forests from further deforestation. Most of the disturbances which occurred during the implementation of the policy were dominated by natural disturbances (e.g., forest fires), and man-made deforestation did not pose a threat.

## 6. Conclusions

Based on the long-term Modis and Landsat time-series, this study explored the impact of major ecological projects on the ecological environment using the SDG index and land disturbance index in GKM. Our study found that the browning trends of vegetation coexisted with tree transition to grass. Only in some areas was there some degree of land productivity degradation, while most other areas showed improvement. Multiple factors such as precipitation, temperature, and the inappropriate afforestation led to a contrasting pattern of vegetation restoration. To remove the effect of precipitation, the effect of precipitation is considered in the calculation of land productivity and its effect is removed in the algorithm. Meanwhile, to identify the causes of the forest disturbance, the LandTrendr algorithm explored several places with high disturbance and analyzed the causes. The land degradation index of SDG 15.3.1 could be used to identify the land type change and changes in land productivity. As discussed in the previous section, land use change as the first indicator of ecological effects provided a direct indication of land change under the influence of the policy.

The study emphasizes that the continuous monitoring and effective management are the keys to the successful implementation of large-scale ecological projects. In addition, in the goal of the ecological restoration project, strengthening the functions of integrated ecosystems should be prioritized over increasing vegetation coverage. We suggest that future research should pay more attention to the scientific planning of large-scale ecological restoration projects, whilst fully considering local conditions and the goal of promoting ecosystem functions.

## Figures and Tables

**Figure 1 ijerph-18-10625-f001:**
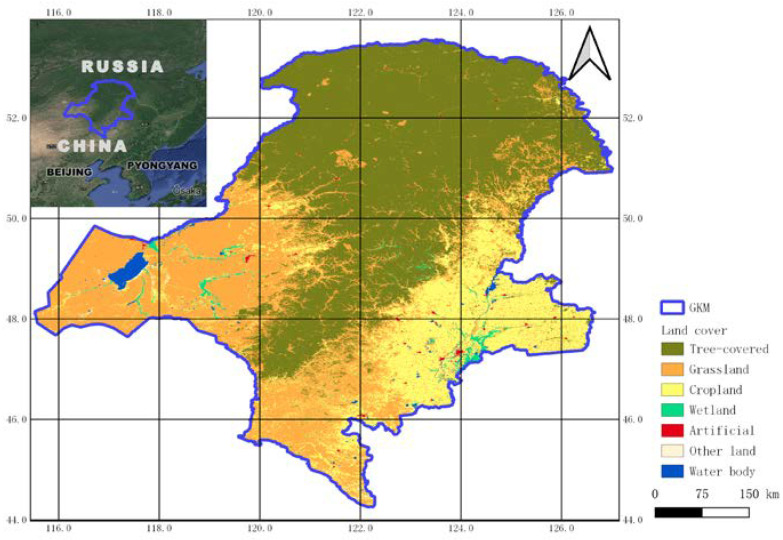
Land use and land cover in Great Khingan Mountain (GKM), China in 2000.

**Figure 2 ijerph-18-10625-f002:**
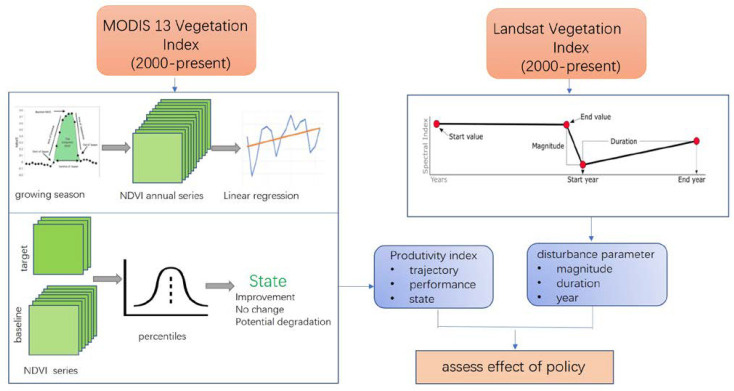
Scheme for calculating land productivity and disturbance for assessment effect of policy in Great Khingan Mountain (GKM), China.

**Figure 3 ijerph-18-10625-f003:**
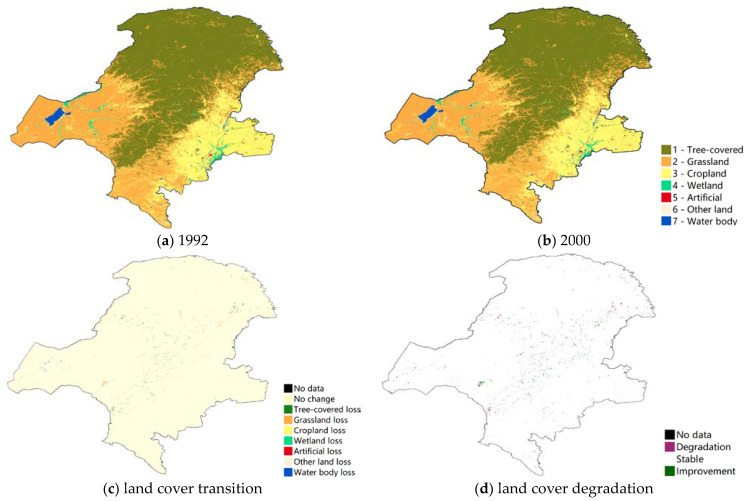
Land cover change and degradation in Great Khingan Mountain (GKM), China (1992–2000).

**Figure 4 ijerph-18-10625-f004:**
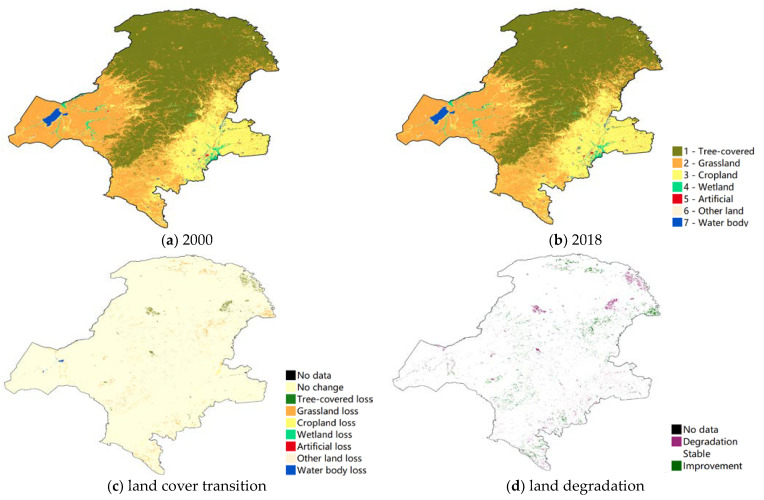
Land cover change and degradation in Great Khingan Mountain (GKM), China (2000–2018).

**Figure 5 ijerph-18-10625-f005:**
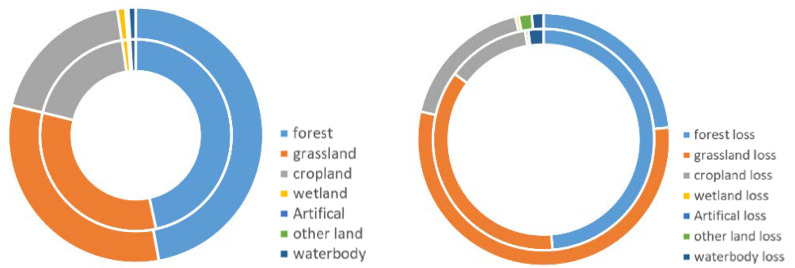
Land cover change and loss from 1992 to 2018 in Great Khingan Mountain (GKM), China (inner circle: 1992–2000; outer circle: 2000–2018).

**Figure 6 ijerph-18-10625-f006:**
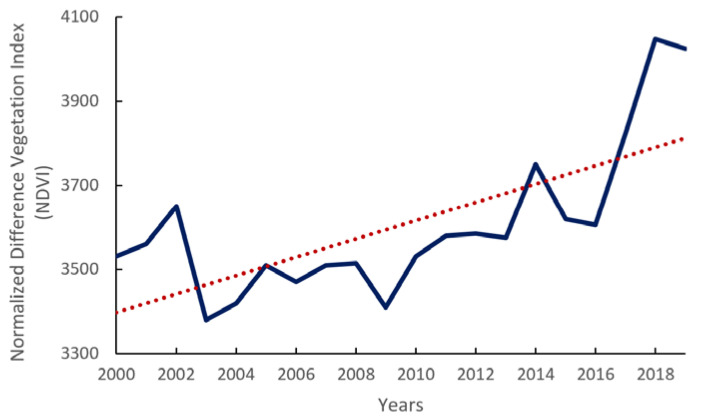
Normalized Difference Vegetation Index (NDVI) trend from 2000 to 2019, in Great Khingan Mountain (GKM), China.

**Figure 7 ijerph-18-10625-f007:**
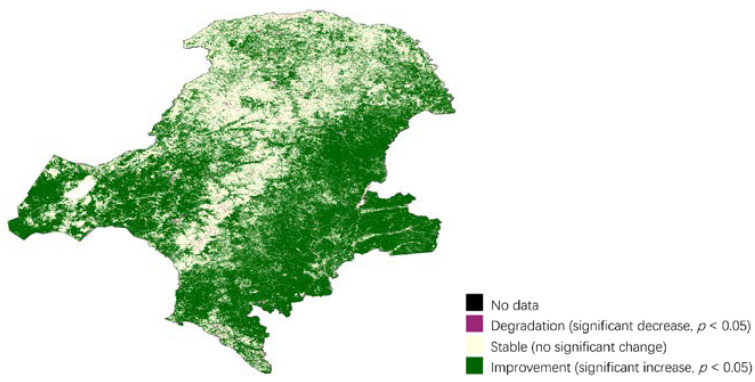
Productivity trajectory degradation in Great Khingan Mountain (GKM), China (2000–2018).

**Figure 8 ijerph-18-10625-f008:**
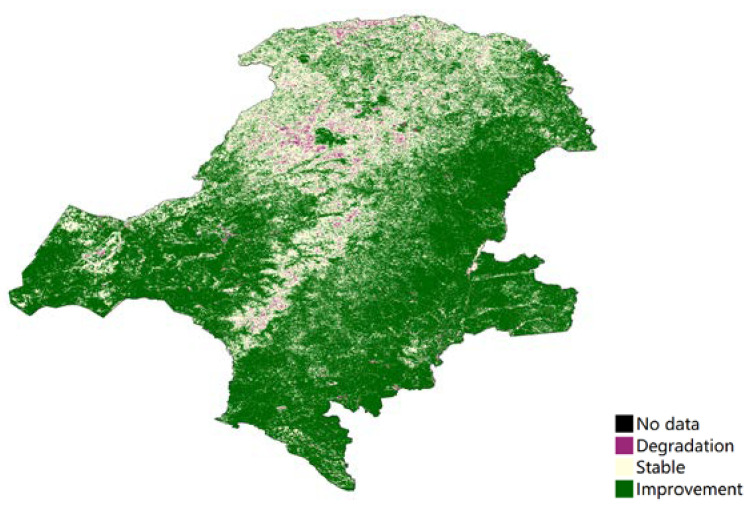
Productivity state degradation in Great Khingan Mountain (GKM), China (2001–2010 vs. 2011–2018).

**Figure 9 ijerph-18-10625-f009:**
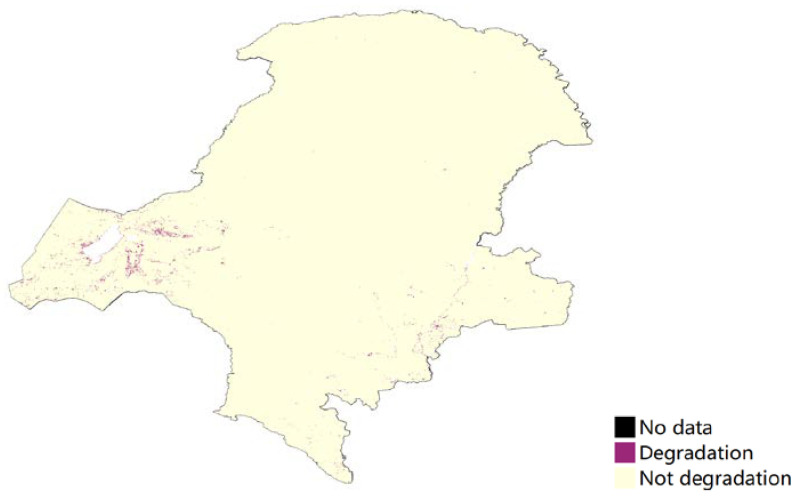
Land productivity performance degradation (2000–2018) in Great Khingan Mountain (GKM), China.

**Figure 10 ijerph-18-10625-f010:**
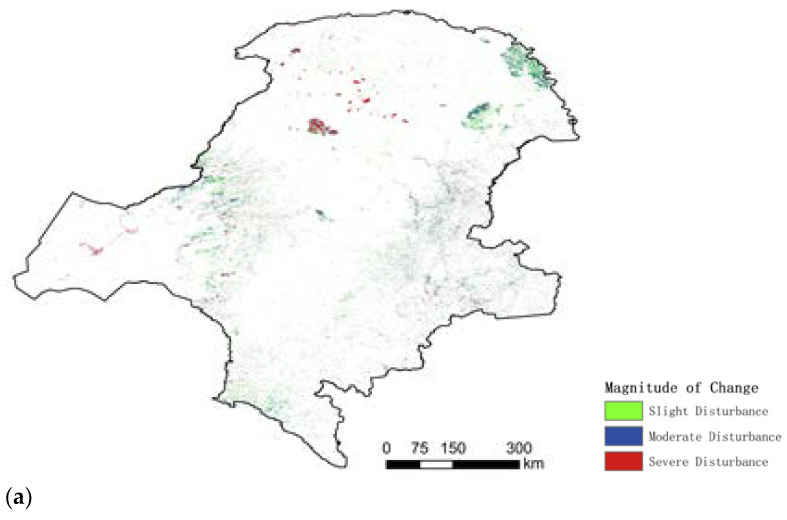
Forest disturbance and recovery at multi-scale and time series in Great Khingan Mountain (GKM), China: (**a**) magnitude of change, (**b**) duration of change, and (**c**) year of detection.

**Figure 11 ijerph-18-10625-f011:**
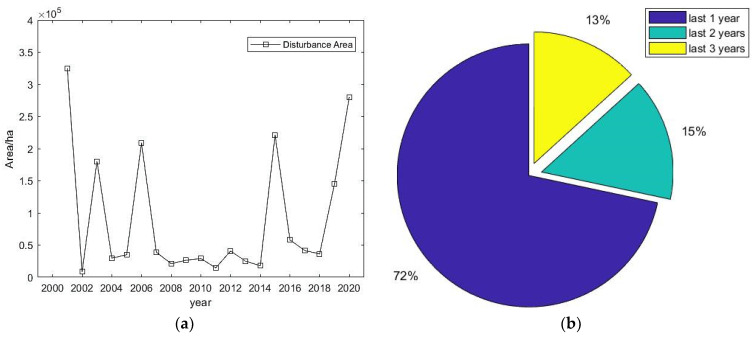
(**a**) Disturbance area and (**b**) duration year, in Great Khingan Mountain (GKM), China.

**Figure 12 ijerph-18-10625-f012:**
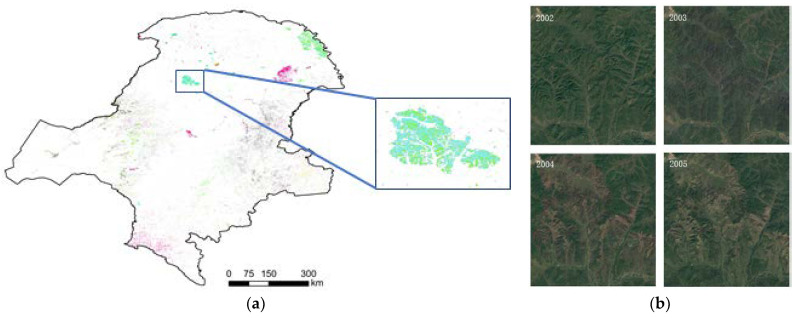
(**a**) Typical disturbance area and (**b**) remote sensing images, in Great Khingan Mountain (GKM), China.

**Figure 13 ijerph-18-10625-f013:**
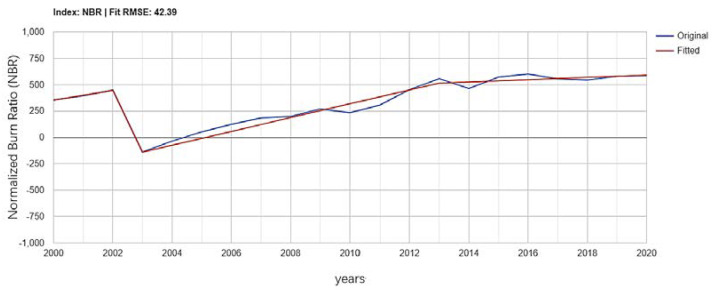
Normalized Burn Ratio (NBR) index trend from 2000 to 2020, in typical disturbance area.

**Table 1 ijerph-18-10625-t001:** Remote sensing image data source for Great Khingan Mountain (GKM), China.

Sensor	Satellite	Frequency	Data Source	Data Record	Spatial Resolution	Time Step
MODIS	Terra/Aqua	1–2 days	MOD13 vegetation index	2000–present	250 m, 500 m, 1 km	8-day, 16-day
TM	Landsat 4–5	16 days	USGS/EROS	1982–2011	30 m	Distributed by scene
ETM+	Landsat 7	16 days	USGS/EROS	1999–presnet	30 m	Distributed by scene
OLI	Landsat 8	16 days	USGS/EROS	2013–present	30 m	Distributed by scene

## Data Availability

Not applicable.

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
