# Peer review of "Ecological Effect of Ecological Engineering Projects on Low-Temperature Forest Cover in Great Khingan Mountain, China"

_ijerph, 2021, doi:10.3390/ijerph182010625_

Round 1

Reviewer 1 Report

The manuscript brings a reflexion about the putative utility of multi-parameters assessment (land cover, land use, climate data) to addresses ecological restoration. The case study was conducted in the Greater Khingan Mountains (China) with two main goals 1) to assess the status of forest disturbance and recovery using multi-scale satellite images; and 2) to infer on the impacts of land use change and the existing forest policies. The manuscript has several problems; presenting the goals; describing the methodology; exposing the results. The discussion does not argue critically the investigation with the literature available.

Author Response

We genuinely appreciate the reviewers’ valuable comments, which are really useful for our paper revision and improvement. Following the comments, we have accepted the comments and suggestions and made the point-by-point revisions. The details are marked in the R4 version with change marked.

Reviewer 1:The manuscript brings a reflexion about the putative utility of multi-parameters assessment (land cover, land use, climate data) to addresses ecological restoration. The case study was conducted in the Greater Khingan Mountains (China) with two main goals 1) to assess the status of forest disturbance and recovery using multi-scale satellite images; and 2) to infer on the impacts of land use change and the existing forest policies. The manuscript has several problems; presenting the goals; describing the methodology; exposing the results. The discussion does not argue critically the investigation with the literature available.

Question: (1) presenting the goals?

Answer (1): We added the hypotheses in the manuscript:

“Forest change can be influenced by many factors. In this paper, the influence of other climatic factors will be removed from the calculations to study the impact of ecological projects, meaning that the study assumes that external factors such as climate are consistent over the period of forest change in the study area.”

 And We have revised the objective as follows:

“This paper will use images at different scales to identify the forest degradation and improvement for exploring the possible impacts in different contexts and policies.”

Question: (2) describing the methodology?

Answer (2): We added the details of data period used and period of ecological program (Natural Forest Protect Project) in section “3 method”.

“As the NFPP was a long-term project, this paper will use 2000 as a cut-off point to analyse the different land use changes and land degradation before and after the NFPP.  The year 2010, the start of the second phase of the NFPP, was chosen as the interval. Early epoch (2001 to 2010) and late epoch (2011 to 2018) average productivity will be calculated. Then, pixels in which the productivity level has dropped between early and late epochs were identified for this metric.”

We corrected the period of Landsat vegetation index consistent with period of MODIS vegetation in Figure 2. Meanwhile, we adjusted the period of annual mean NDVI to be better matched with the period of land use changes, recalculated, and redraw the trend of NDVI in Figure 6.

Question: (3) exposing the results

Answer (3): We redraw all the figures in the results and improved their quality. We added the quantitative data to compare the land productivity degradation in section 4.1. Then, we removed some sentences about the methodology.

Question: (4) The discussion does not argue critically the investigation with the literature available.

Answer (4): We reorganized the Discussion 5.1 and 5.2. To increase the reliability of the discussion, we have also added 13 references in the manuscript.

Reviewer 2 Report

Article Review: Ecological effect of ecological engineering projects on low-temperature forest cover in Great Khingan Mountain, China

The article has scientific relevance and aims to investigate the dynamics of environmental degradation processes in forest areas and their respective recovery processes in a vast area in northern China.

However, it needs important adjustments, including making it clear to the reader from the outset the real period of its analysis. There is confusion about this. Does Figure 2 show that there is MODIS NDVI data from 2000 to the present, or was this entire period used to verify vegetation dynamics? Similarly in the same figure it shows vegetation index products from 1990 to the present LANDSAT, which LANDSAT specifically? Is this the period of data provided by the satellite or used in your study? These details need to be made clearer to the reader.

General observations:

1- Adapt the references to the journal's standard;

2- Improve the resolution of Figure 1;

2.2.1. Evidence of used NDVI data period;

2.2.1. Evidence of the used CHIRPS data period;

3.1. It needs to make clearer what the iET index of equation 1 means. The Mann Kendall technique is mentioned for trend detection with just one reference. Use more recent, such as:

Costa, R.L.; Baptista, G.M.M.; Gomes, H.B.; Silva, F.D.S.; da Rocha Júnior, R.L.; Salvador, M.A.; Herdies, D.L. Analysis of climate extremes indices over northeast Brazil from 1961 to 2014. Weather and Climate Extremes, v. 28, p. 100254, 2020.

4.1. According to Figures 3 and 4, their study was continued over time, or was based on a punctual comparison of years: 1992 and 2000; 2000 and 2018? The visual quality of the maps of these figures needs to be considerably improved. The NDVI trend is for a period different from the first comparative analysis, can't you standardize the analysis periods? Do not use topics separated by numbers, (1); (2), create subtopics or follow in a continuous text structure. Figure 7 again shows an important result, but for a different period in relation to the previous ones. Justify why this happens.

4.2. Figure 10 brings important results related to changes in vegetation dynamics, but they are illegible.

  1. Discussion without referencing other similar studies is not discussion, it is just an extension of the description of your results. It is necessary to compare their results to other studies, even if they are from other regions of the globe, if there are no similar studies for China to support, in this case, a real discussion.
  2. (C)onclusion(s) or Conclusion or Final considerations – correct because “conclusion” appears with a lowercase first letter, and must be plural because more than one conclusion is found.

Author Response

We genuinely appreciate the reviewers’ valuable comments, which are really useful for our paper revision and improvement. Following the comments, we have accepted the comments and suggestions and made the point-by-point revisions. The details are marked in the R4 version with change marked.

Reviewer 2:The article has scientific relevance and aims to investigate the dynamics of environmental degradation processes in forest areas and their respective recovery processes in a vast area in northern China.

However, it needs important adjustments, including making it clear to the reader from the outset the real period of its analysis. There is confusion about this. Does Figure 2 show that there is MODIS NDVI data from 2000 to the present, or was this entire period used to verify vegetation dynamics? Similarly in the same figure it shows vegetation index products from 1990 to the present LANDSAT, which LANDSAT specifically? Is this the period of data provided by the satellite or used in your study? These details need to be made clearer to the reader.

Answer: We added the details of data period used and period of ecological program (Natural Forest Protect Project) in section “3 method”.

“As the NFPP was a long-term project, this paper will use 2000 as a cut-off point to analyse the different land use changes and land degradation before and after the NFPP.  The year 2010, the start of the second phase of the NFPP, was chosen as the interval. Early epoch (2001 to 2010) and late epoch (2011 to 2018) average productivity will be calculated. Then, pixels in which the productivity level has dropped between early and late epochs were identified for this metric.”

We corrected the period of Landsat vegetation index consistent with period of MODIS vegetation in Figure 2. Meanwhile, we adjusted the period of annual mean NDVI to be better matched with the period of land use changes, recalculated, and redraw the trend of NDVI in Figure 6.

General observations:

1- Adapt the references to the journal's standard;

Answer: We revised the format of the references.

2- Improve the resolution of Figure 1;
Answer: We redraw and improved the resolution of Figure 1.

2.2.1. Evidence of used NDVI data period;

Answer: We used two types of NDVI data. One was MODIS (MOD13 Q1) started from 2000 to near present. The other NDVI data was Landsat data started from 1982 to present. Because the period of NFPP was starting from 2000 to 2020, we selected the NDVI data from 2000. And we revised the mistake in period of NDVI data in Figure 2 and 6.

2.2.1. Evidence of the used CHIRPS data period;

Answer: The period of CHIRPS data was starting from 1981 to the near present. We used CHIRPS for computing the land productivity from 2000 in GKM. So, the period of CHIRPS is from 2000 to the near present.

3.1. It needs to make clearer what the iET index of equation 1 means. The Mann Kendall technique is mentioned for trend detection with just one reference. Use more recent, such as:

Costa, R.L.; Baptista, G.M.M.; Gomes, H.B.; Silva, F.D.S.; da Rocha Júnior, R.L.; Salvador, M.A.; Herdies, D.L. Analysis of climate extremes indices over northeast Brazil from 1961 to 2014. Weather and Climate Extremes, v. 28, p. 100254, 2020.

Answer: ET was “Evapotranspiration”. We added it in the manuscript. And we added more reference about the Mann Kendall technique.

“Costa, R.L.; Baptista, G.M.M.; Gomes, H.B.; Silva, F.D.S.; da Rocha Júnior, R.L.; Salvador, M.A.; Herdies, D.L. Analysis of climate extremes indices over northeast Brazil from 1961 to 2014. Weather and Climate Extremes, v. 28, p. 100254, 2020.”

4.1. According to Figures 3 and 4, their study was continued over time, or was based on a punctual comparison of years: 1992 and 2000; 2000 and 2018? The visual quality of the maps of these figures needs to be considerably improved. The NDVI trend is for a period different from the first comparative analysis, can't you standardize the analysis periods? Do not use topics separated by numbers, (1); (2), create subtopics or follow in a continuous text structure. Figure 7 again shows an important result, but for a different period in relation to the previous ones. Justify why this happens.

Answer: (1). Because the NFPP project began from 2000, we selected 2000 as a base year to compare the land use changes before and after the NFPP considering the availability of ESA CCI data. We got the land use map each year from 1992 to 2018 and showed 1992, 2000, and 2018 only. We redraw the Figure 3 and 4.

(2). We revised the period of the NDVI trend consistent with the period of MODIS data (from 2000 to 2018) in Figure 6.  

(3) We removed the subtopics in section 4.1. And we added the period of land productivity (from 2000 to 2018) to identify the degradation or improvement of land productivity after the implementation of NFPP.

4.2. Figure 10 brings important results related to changes in vegetation dynamics, but they are illegible.

Answer: We improved the quality of Figure 10.

  1. Discussion without referencing other similar studies is not discussion, it is just an extension of the description of your results. It is necessary to compare their results to other studies, even if they are from other regions of the globe, if there are no similar studies for China to support, in this case, a real discussion.

Answer: We reorganized the Discussion 5.1 and 5.2. To increase the reliability of the discussion, we have also added 13 references in the manuscript.

  1. (C)onclusion(s) or Conclusion or Final considerations – correct because “conclusion” appears with a lowercase first letter, and must be plural because more than one conclusion is found.

Answer: We revised “Conclusions” in the manuscript. 

Reviewer 3 Report

General comments to authors:

The manuscript "Ecological effect of ecological engineering projects on low-temperature forest cover in Great Khingan Mountain, China" has two main objectives: (1) to use images at different scales to identify forest degradation and improvement; (2) to explore the possible impacts in different contexts and interrelate them with existing forest policies.

This research presents the results of considerable sampling effort on the part of the authors.

However, in general, some topics need improvement: (i) The introduction is partially clear. The literature review needs to help set the stage for the rest of the article, so some topics need to be clarified (see detailed comments). The study lacks clearly stated hypotheses; (ii) The analysis was thorough and well documented; however, the methods lack a better explanation (see detailed comments); (iii) Results were clearly presented; but (iv) I feel that a Discussion section with the literature can make the article better (several works were cited in the introduction and other sections and should be used to discuss the results). The article discussion is extremely fragile.

Detailed comments to authors:

I inserted the considerations below and in the attached PDF.

Abstract:

Point 1; Line 14: Normalized Difference Vegetation Index

Introduction:

General: Several sentences need space (I've highlighted some in red).

Point 1; Line 29: remove

Point 2; Line 29: And the first?

Point 3; Line 32: What is the concept?

Point 4; Lines 38-39: something confusing in writing.

Point 5; Lines 106-107: Authors can write as a general objective only!

Point 6; Lines 105-108: The study lacks clearly stated hypotheses.

Materials and Method:

Point 1; Line 124: Very vague caption, it should be self-explanatory.

Point 2; Line 183: What is?

Point 3; Lines 201-205: Show it in your results.

Results:

Point 1; Lines 2018-2019: Methodology!

Point 2; Figure 3: Improve the quality.

Point 3; Figure 4: Improve the quality.

Point 4; Figure 6: Improve the quality.

Point 5; Line 292: And the productivity classes?

Point 6; Figures 7 and 8: Improve the quality.

Point 7; Lines 301-304: Methodology

Point 8; Lines 307-309: Methodology

Point 9; Figure 9: Improve the quality.

Point 10; Lines 326-327: Or restauration effect?

Point 11; Figure 10: Improve the quality.

Point 12; Line 343: Quantitative data like this might be better for your article compared to your images.

Point 13; Figure 11: Improve the quality.

Point 14; Lines 364-365: Inform in the methodology.

Point 15; Figure 12: Improve the quality.

Point 16; Line 388: Review the beginning of the sentence.

Point 17; Figure 13: Improve the quality.

Discussion:

General: This section seems to me a historical rescue. It must have a discussion face! He should also discuss with the specialized scientific literature.

Point 1; Lines 403-426: Source?

Point 2; Lines 403-426: Source?

Point 3; Lines 428-443: Idem.

Point 4; Lines 444-455: Source? Your study?

Conclusion:

Point 1; Line 491: You can use this goal to discuss.

Author Response

We genuinely appreciate the reviewers’ valuable comments, which are really useful for our paper revision and improvement. Following the comments, we have accepted the comments and suggestions and made the point-by-point revisions. The details are marked in the R4 version with change marked.

Reviewer 3:The manuscript "Ecological effect of ecological engineering projects on low-temperature forest cover in Great Khingan Mountain, China" has two main objectives: (1) to use images at different scales to identify forest degradation and improvement; (2) to explore the possible impacts in different contexts and interrelate them with existing forest policies.

This research presents the results of considerable sampling effort on the part of the authors.

However, in general, some topics need improvement: (i) The introduction is partially clear. The literature review needs to help set the stage for the rest of the article, so some topics need to be clarified (see detailed comments). The study lacks clearly stated hypotheses; (ii) The analysis was thorough and well documented; however, the methods lack a better explanation (see detailed comments); (iii) Results were clearly presented; but (iv) I feel that a Discussion section with the literature can make the article better (several works were cited in the introduction and other sections and should be used to discuss the results). The article discussion is extremely fragile.

Detailed comments to authors:

I inserted the considerations below and in the attached PDF.

Abstract:

Point 1; Line 14: Normalized Difference Vegetation Index (Revised)

Answer: We have added it in the paper.

Introduction:

General: Several sentences need space (I've highlighted some in red). (Revised)

Answer: We have added space.

Point 1; Line 29: remove (Revised)

Answer: We have removed “ecological”.

Point 2; Line 29: And the first? (Revised)

Answer: “The first and the second wave” has no clean time frame, and we revised it for “ a new wave”.

Point 3; Line 32: What is the concept? (Revised)

Answer: The concept of “sustainable ecological development” was not very clear and we revised for “environmental sustainability”.

Point 4; Lines 38-39: something confusing in writing. (Revised)

Answer:  We have revised this sentence as follows:

“Reducing Emissions from Deforestation and Forest Degradation plus (REDD+), as the best-known international forestry-based policy for carbon dioxide removal, was created by the United Nations Framework Convention on Climage Change (UNFCCC) Conference and aimed at the implementation of activities by national governments to reduce human pressure on forests that result in greenhouse gas emissions at the national level.”

Point 5; Lines 106-107: Authors can write as a general objective only!

Answer:  We have revised the objective as follows:

“this paper will use images at different scales to identify the forest degradation and improvement for exploring the possible impacts in different contexts and policies.”

Point 6; Lines 105-108: The study lacks clearly stated hypotheses.

Answer:  We added the hypotheses in the manuscript:

“Forest change can be influenced by many factors. In this paper, the influence of other climatic factors will be removed from the calculations to study the impact of ecological projects, meaning that the study assumes that external factors such as climate are consistent over the period of forest change in the study area.”

Materials and Method:

Point 1; Line 124: Very vague caption, it should be self-explanatory.

Answer: We revised the caption of Figure. 1  “Figure. 1 Land use and land cover in GKM in 2000”

Point 2; Line 183: What is?

Answer: ET is Evapotranspiration. We added it in the manuscript and added the reference.

Point 3; Lines 201-205: Show it in your results.

Answer: We added it in the results in Section “4.1 (3) land productivity”

Results:

Point 1; Lines 2018-2019: Methodology!

Answer: We removed this sentence from results and put it in the section “3.1. Land productivity”.

Point 2; Figure 3: Improve the quality.

Point 3; Figure 4: Improve the quality.

Point 4; Figure 6: Improve the quality.

Answer: We redraw the Figure 3, 4, 5 and improved their quality.

Point 5; Line 292: And the productivity classes?   ??

Answer: We remove the subheading of “Land productivity”. There are three types of land productivity degradation: state degradation, trajectory degradation, and performance degradation.

Point 6; Figures 7 and 8: Improve the quality.

Answer: We redraw the Figure 7, 8 and improved their quality.

Point 7; Lines 301-304: Methodology

Answer: We removed this sentence from results and put it in the section “3.1. Land productivity”.

“As the NFPP was a long-term project, this paper will use 2000 as a cut-off point to analyse the different land use changes and land degradation before and after the NFPP. The year 2010, the start of the second phase of the NFPP, was chosen as the interval. Early epoch (2001 to 2010) and late epoch (2011 to 2018) average productivity will be calculated. Then, pixels in which the productivity level has dropped between early and late epochs were identified for this metric.”

Point 8; Lines 307-309: Methodology

Answer: We removed this sentence from results and put it in the section “3.1. Land productivity”.

Point 9; Figure 9: Improve the quality.

Answer: We redraw the Figure 9 and improved their quality.

Point 10; Lines 326-327: Or restauration effect?   ??

Answer: The Landtrendr algorithm used annual NDVI data to identify disturbances. The GKM region was located in an alpine area and it was very difficult to recover in a short time if disturbances happened, so if no disturbance was identified in this area, it means that the forest was maintaining a good condition.

Point 11; Figure 10: Improve the quality.

Answer: We redraw the Figure 10 and improved their quality.

Point 12; Line 343: Quantitative data like this might be better for your article compared to your images.

Answer: We added the quantitative data to compare the land productivity degradation in section 4.1.

Point 13; Figure 11: Improve the quality.

Answer: We redraw the Figure 11 and improved their quality.

Point 14; Lines 364-365: Inform in the methodology.

Answer: We put it in the section “3.2. LandTrendr”.

“Meanwhile, the typical area will be selected and compared with the Landsat remote sensing images selected on Google Earth to explore the impossible reason of disturbance.”

Point 15; Figure 12: Improve the quality.

Answer: We redraw the Figure 12 and improved their quality.

Point 16; Line 388: Review the beginning of the sentence.

Answer: We revised this sentence.

Point 17; Figure 13: Improve the quality.

Answer: We redraw the Figure 13 and improved their quality.

Discussion:

General: This section seems to me a historical rescue. It must have a discussion face! He should also discuss with the specialized scientific literature.

Answer: We reorganized the Discussion 5.1 and 5.2. To increase the reliability of the discussion, we have also added 13 references in the manuscript.

Point 1; Lines 403-426: Source?

Point 2; Lines 403-426: Source?

Point 3; Lines 428-443: Idem.

Point 4; Lines 444-455: Source? Your study?

Answer: We added the references in the manuscript in lines 403-426, 428-443, 444-445 to confirm our results and discuss.

Conclusion:

Point 1; Line 491: You can use this goal to discuss.

Answer: Yes, we added the SDG to the discuss 5.2.

“This paper used SDG 15.3.1 as an important indicator for the evaluation the effectiveness of large-scale ecological projects, with three sub-objectives in land productivity trajectory degradation, state degradation, performance degradation. The results were consistent with the actual situation such as actual land use change and policy direction. The sub-indicator based on SDG 15.3.1 will provide a reliable and practical method to assess the effect of ecological projects.”

Round 2

Reviewer 1 Report

The manuscript was significantly improved. However, there are still inaccuracies regarding the study design in terms of the criteria versus difference(s) between study areas / study sites / treatments /control.

Other imprecisions that should be corrected:

-Abstract (line 11) “Aiming at the current problem of single evaluation index, this article will evaluate the implementation effect of ecological projects from multiple dimensions”; please specify the dimensions approached.

- Many times it is used the expression ‘this paper’; it should be carefully checked and the terms ‘research’, ‘study design’, ‘study’ adopted preferably.

-Several phrases are in the future. Text should be carefully checked; it is about past research and past approach.

Author Response

We genuinely appreciate your valuable comments. Following the comments, we have accepted the comments and suggestions and made the point-by-point revisions. The details are marked in the R5 version with change marked.

The manuscript was significantly improved. However, there are still inaccuracies regarding the study design in terms of the criteria versus difference(s) between study areas / study sites / treatments /control.

Answer: There are some difficulties in the design and implementation of the process due to the periodicity and consistency of data acquisition. So, we divided three time point to evaluate the effect of the NFPP: before the NFPP (before 2000), the first phase of the NFPP (2000-2010) and the second phase of the NFPP (2010-present). Meanwhile, different index have used in different phases.

Other imprecisions that should be corrected:

-Abstract (line 11) “Aiming at the current problem of single evaluation index, this article will evaluate the implementation effect of ecological projects from multiple dimensions”; please specify the dimensions approached.

Answer: We revised this sentence as shown below:

“Aiming at the current problem of single evaluation index, this study evaluated the implementation effect of ecological projects from different temporal and spatial dimensions.”

- Many times it is used the expression ‘this paper’; it should be carefully checked and the terms ‘research’, ‘study design’, ‘study’ adopted preferably.

Answer: We revised “this paper” to “study” or “research”.

-Several phrases are in the future. Text should be carefully checked; it is about past research and past approach.

Answer: We changed the tense of some sentences from the future tense to the past tense.

Reviewer 2 Report

All my questions have been satisfactorily answered. My suggestions were accepted.

Author Response

We genuinely appreciate your valuable comments, which are really useful for our paper revision and improvement. Thank you very much.

Reviewer 3 Report

The manuscript "Ecological effect of ecological engineering projects on low-temperature forest cover in Great Khingan Mountain, China" has two main objectives: (1) to use images at different scales to identify forest degradation and improvement; (2) to explore the possible impacts in different contexts and interrelate them with existing forest policies. This research presents the results of considerable sampling effort on the part of the authors. Overall, the provided information is a good resource. The authors have made efforts to amend the manuscript based upon the original comments. The changes made by the authors improve the manuscript. In short, I believe this manuscript, after minor revisions, is suitable for publication.

Some details:

Line 63: Insert space between: productivity (NPP)

Line 88: Great Khingan Mountain (GKM)

Line 118: 2. Study Area and Data

Line 119: 2.1. Study area

Line 133: Figure 1. Land use and land cover in Great Khingan Mountain (GKM), China in 2000.

Lines 134: 2.2. Data

Line 148: Table 1. Remote sensing image data source in Great Khingan Mountain (GKM), China.

Line 149: 2.2.2. Land cover and climate data

Line 187: Figure 2. Scheme for calculating land productivity and disturbance for assessment effect of policy in Great Khingan Mountain (GKM), China.

Line 243: 4.1. Trend and changes of land productivity index

Line 281: Figure 3. Land cover change and degradation in Great Khingan Mountain (GKM), China (1992-2000).

Line 303: Figure 4. Land cover change and degradation in Great Khingan Mountain (GKM), China (2000-2018).

Line 320: Figure 5. Land cover change and loss from 1992 to 2018 in Great Khingan Mountain (GKM), China (inner circle: 1992-2000; outer circle: 2000-2018).

Line 330: Figure 6. Normalized Difference Vegetation Index (NDVI) trend from 2000 to 2019, in Great Khingan Mountain (GKM), China.

Figure 6: Remove the grids from the figure. Leave only the X and Y Axes.

Enter title on the axes. X axis: Years; Y axis: Normalized Difference Vegetation Index (NDVI). In this way, remove "NDVI" as a figure title.

Figures 7 and 8: Move figure 8 below figure 7.

Line 348: Figure 7. Productivity trajectory degradation in Great Khingan Mountain (GKM), China (2000-2018).

Line 348: Figure 8. Productivity state degradation in Great Khingan Mountain (GKM), China (2001-2010 vs 2011-2018).

Line 371: Figure 9. Land productivity performance degradation (2000-2018) in Great Khingan Mountain (GKM), China.

Line 433: Figure 10. Forest disturbance and recovery at multi-scale and time series in Great Khingan Mountain (GKM), China: (a) magnitude of change, (b) duration of change, and (c) year of detection.

Line 454: Figure 11. (a) Disturbance area and (b) duration year, in Great Khingan Mountain (GKM), China.

Line 467: NBR is the Normalized Burn Ratio (NBR)?)

Line 484: Figure 12. (a) Typical disturbance area and (b) remote sensing images, in Great Khingan Mountain (GKM), China.

Figure 13: Enter title on the axes. X axis: Years; Y axis: Normalized Burn Ratio (NBR).

Line 489: Review the caption for this figure. I think she is not correct.

Line 491: 5.1. Land use change trend

Line 515: 5.2. Land productivity trend

Author Response

We genuinely appreciate your valuable comments. Following the comments, we have accepted the comments and suggestions and made the point-by-point revisions. The details are marked in the R5 version with change marked.

The manuscript "Ecological effect of ecological engineering projects on low-temperature forest cover in Great Khingan Mountain, China" has two main objectives: (1) to use images at different scales to identify forest degradation and improvement; (2) to explore the possible impacts in different contexts and interrelate them with existing forest policies. This research presents the results of considerable sampling effort on the part of the authors. Overall, the provided information is a good resource. The authors have made efforts to amend the manuscript based upon the original comments. The changes made by the authors improve the manuscript. In short, I believe this manuscript, after minor revisions, is suitable for publication.

Some details:

Line 63: Insert space between: productivity (NPP)

Answer: We have added space.

Line 88: Great Khingan Mountain (GKM)

Answer: We have added Great Khingan Mountain before GKM.

Line 118: 2. Study Area and Data

Answer: We have revised this caption.

Line 119: 2.1. Study area

Answer: We have revised this caption.

Line 133: Figure 1. Land use and land cover in Great Khingan Mountain (GKM), China in 2000.

Answer: We have revised this caption.

Lines 134: 2.2. Data

Answer: We have revised this caption.

Line 148: Table 1. Remote sensing image data source in Great Khingan Mountain (GKM), China.

Answer: We have revised this caption.

Line 149: 2.2.2. Land cover and climate data

Answer: We have revised this caption.

Line 187: Figure 2. Scheme for calculating land productivity and disturbance for assessment effect of policy in Great Khingan Mountain (GKM), China.

Answer: We have revised this caption.

Line 243: 4.1. Trend and changes of land productivity index

Answer: We have revised this caption.

Line 281: Figure 3. Land cover change and degradation in Great Khingan Mountain (GKM), China (1992-2000).

Answer: We have revised this caption.

Line 303: Figure 4. Land cover change and degradation in Great Khingan Mountain (GKM), China (2000-2018).

Answer: We have revised this caption.

Line 320: Figure 5. Land cover change and loss from 1992 to 2018 in Great Khingan Mountain (GKM), China (inner circle: 1992-2000; outer circle: 2000-2018).

Answer: We have revised this caption.

Line 330: Figure 6. Normalized Difference Vegetation Index (NDVI) trend from 2000 to 2019, in Great Khingan Mountain (GKM), China.

Answer: We have revised this caption.

Figure 6: Remove the grids from the figure. Leave only the X and Y Axes.

Enter title on the axes. X axis: Years; Y axis: Normalized Difference Vegetation Index (NDVI). In this way, remove "NDVI" as a figure title.

Answer: We have redrawn the Figure. 6.

Figures 7 and 8: Move figure 8 below figure 7.

Answer: We moved Figure 8 below Figure 7.

Line 348: Figure 7. Productivity trajectory degradation in Great Khingan Mountain (GKM), China (2000-2018).

Answer: We have revised this caption.

Line 348: Figure 8. Productivity state degradation in Great Khingan Mountain (GKM), China (2001-2010 vs 2011-2018).

Answer: We have revised this caption.

Line 371: Figure 9. Land productivity performance degradation (2000-2018) in Great Khingan Mountain (GKM), China.

Answer: We have revised this caption.

Line 433: Figure 10. Forest disturbance and recovery at multi-scale and time series in Great Khingan Mountain (GKM), China: (a) magnitude of change, (b) duration of change, and (c) year of detection.

Answer: We have revised this caption.

Line 454: Figure 11. (a) Disturbance area and (b) duration year, in Great Khingan Mountain (GKM), China.

Answer: We have revised this caption.

Line 467: NBR is the Normalized Burn Ratio (NBR)?)

Answer: Yes, NBR is the Normalized Burn Ratio and we added it in this paper.

Line 484: Figure 12. (a) Typical disturbance area and (b) remote sensing images, in Great Khingan Mountain (GKM), China.

Answer: We have revised this caption.

Figure 13: Enter title on the axes. X axis: Years; Y axis: Normalized Burn Ratio (NBR).

Answer: We added the title of X and Y axis in Figure 13.

Line 489: Review the caption for this figure. I think she is not correct.

Answer: We revised the caption of Figure 13.

“Figure 13. Normalized Burn Ratio (NBR) index trend from 2000 to 2020, in typical disturbance area.”

Line 491: 5.1. Land use change trend

Answer: We have revised this caption.

Line 515: 5.2. Land productivity trend

Answer: We have revised this caption.